# EvoNet: A Neural Network for Predicting the Evolution of Dynamic Graphs

## Abstract

Neural networks for structured data like graphs have been studied extensively in recent years. To date, the bulk of research activity has focused mainly on static graphs. However, most real-world networks are dynamic since their topology tends to change over time. Predicting the evolution of dynamic graphs is a task of high significance in the area of graph mining. Despite its practical importance, the task has not been explored in depth so far, mainly due to its challenging nature. In this paper, we propose a model that predicts the evolution of dynamic graphs. Specifically, we use a graph neural network along with a recurrent architecture to capture the temporal evolution patterns of dynamic graphs. Then, we employ a generative model which predicts the topology of the graph at the next time step and constructs a graph instance that corresponds to that topology. We evaluate the proposed model on several artificial datasets following common network evolving dynamics, as well as on real-world datasets. Results demonstrate the effectiveness of the proposed model.

## 1 Introduction

Graph neural networks (GNNs) have emerged in recent years as an effective tool for analyzing graph-structured data (Scarselli et al., 2008; Gilmer et al., 2017; Zhou et al., 2018; Wu et al., 2019). These architectures bring the expressive power of deep learning into non-Euclidean data such as graphs, and have demonstrated convincing performance in several graph mining tasks, including graph classification (Morris et al., 2019), link prediction (Zhang & Chen, 2018), and community detection (Bruna & Li, 2017; Chen et al., 2017). So far, GNNs have been mainly applied to tasks that involve static graphs. However, most real-world networks are dynamic, i.e., nodes and edges are added and removed over time. Despite the success of GNNs in various applications, it is still not clear if these models are useful for learning from dynamic graphs. Although some models have been applied to this type of data, most studies have focused on predicting a low-dimensional representation (i.e., embedding) of the graph for the next time step (Li et al., 2016; 2017; Nguyen et al., 2018; Goyal et al., 2018; Seo et al., 2018; Pareja et al., 2019). These representations can then be used in downstream tasks (Li et al., 2016; Goyal et al., 2018; Meng et al., 2018; Pareja et al., 2019). Predicting the topology of the graph is a task that has not been properly addressed yet.

Graph generation, another important task in graph mining, has attracted a lot of attention from the deep learning community in recent years. The objective of this task is to generate graphs that exhibit specific properties, e.g., degree distribution, node triangle participation, community structure etc. Traditionally, graphs are generated based on some network generation model such as the Erdős-Rényi model. These models focus on modeling one or more network properties, and neglect the others. Neural network approaches, on the other hand, can better capture the properties of graphs since they follow a supervised approach (You et al., 2018; Bojchevski et al., 2018; Grover et al., 2018). These architectures minimize a loss function such as the reconstruction error of the adjacency matrix or the value of a graph comparison algorithm.

Capitalizing on recent developments in neural networks for graph-structured data and graph generation, we propose in this paper, to the best of our knowledge, the first framework for predicting the evolution of the topology of networks in time. The proposed framework can be viewed as an encoder-decoder architecture. The "encoder" network takes a sequence of graphs as input and uses a GNN to produce a low-dimensional representation for each one of these graphs. These representations capture

structural information about the input graphs. Then, it employs a recurrent architecture which predicts a representation for the future instance of the graph. The "decoder" network corresponds to a graph generation model which utilizes the predicted representation, and generates the topology of the graph for the next time step. The proposed model is evaluated over a series of experiments on synthetic and real-world datasets. To measure its effectiveness, the generated graphs need to be compared with the corresponding ground-truth graph instances. To this end, we use the Weisfeiler-Lehman subtree kernel which scales to very large graphs and has achieved state-of-the-art results on many graph datasets (Shervashidze et al., 2011). The proposed model is compared against several baseline methods. Results show that the proposed model is very competitive, and in most cases, outperforms the competing methods.

The rest of this paper is organized as follows. Section 2 provides an overview of the related work and elaborates our contribution. Section 3 introduces some preliminary concepts and definitions related to the graph generation problem, followed by a detailed presentation of the components of the proposed model. Section 4 evaluates the proposed model on several tasks. Finally, Section 5 concludes.

## 2 RELATED WORK

Our work is related to random graph models. These models are very popular in graph theory and network science. The Erdős-Rényi model (Erdős & Rényi, 1960), the preferential attachment model (Albert & Barabási, 2002), and the Kronecker graph model (Leskovec et al., 2010) are some typical examples of such models. To predict how a graph structure will evolve over time, the values of the parameters of these models can be estimated based on the corresponding values of the observed graph instances, and then the estimated values can be passed on to these models to generate graphs.

Other work along a similar direction includes neural network models which combine GNNs with RNNs (Seo et al., 2018; Manessi et al., 2017; Pareja et al., 2019). These models use GNNs to extract features from a graph and RNNs for sequence learning from the extracted features. Other similar approaches do not use GNNs, but they instead perform random walks or employ deep autoencoders (Nguyen et al., 2018; Goyal et al., 2018). All these works focus on predicting how the node representations or the graph representations will evolve over time. However, some applications require predicting the topology of the graph, and not just its low-dimensional representation. The proposed model constitutes the first step towards this objective.

## 3 EVONET: A NEURAL NETWORK FOR PREDICTING GRAPH EVOLUTION

In this Section, we first introduce basic concepts from graph theory, and define our notation. We then present EvoNet, the proposed framework for predicting the evolution of graphs. Since the proposed model comprises of several components, we describe all these components in detail.

### 3.1 PRELIMINARIES

Let $\mathcal{G} = (V, E)$ be an undirected, unweighted graph, where $V$ is the set of nodes and $E$ is the set of edges. We will denote by $n$ the number of vertices and by $m$ the number of edges. We define a permutation of the nodes of $\mathcal{G}$ as a bijective function $\pi : V \to V$, under which any graph property of $G$ should be invariant. We are interested in the topology of a graph which is described by its adjacency matrix $A^\pi \in \mathbb{R}^{n \times n}$ with a specific ordering of the nodes $\pi$[1]. Each entry of the adjacency matrix is defined as $A^\pi_{ij} = \mathbb{1}_{(\pi(v_i), \pi(v_j)) \in E}$ where $v_i, v_j \in V$. In what follows, we consider the "topology", "structure" and "adjacency matrix" of a graph equivalent to each other.

In many real-world networks, besides the adjacency matrix that encodes connectivity information, nodes and/or edges are annotated with a feature vector, which we denote as $X \in \mathbb{R}^{n \times d}$ and $L \in \mathbb{R}^{m \times d}$, respectively. Hence, a graph object can be also written in the form of a triplet $\mathcal{G} = (A, X, L)$. In this paper, we use this triplet to represent all graphs. If a graph does not contain node/edge attributes, we assign attributes to it based on local properties (e. g., degree, $k$-core number, number of triangles, etc).

---

[1]For simplicity, the ordering $\pi$ will be omitted in what follows.

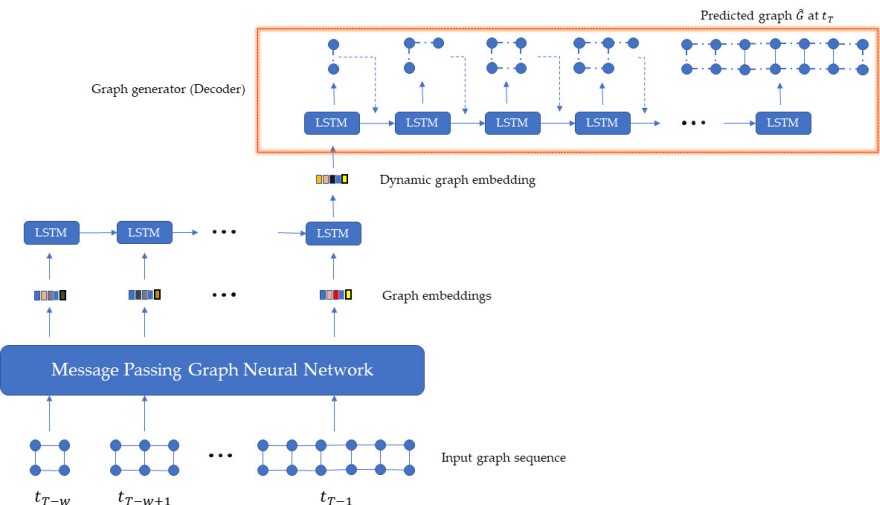

Figure 1: Illustration of the proposed architecture

An evolving network is a graph whose topology changes as a function of time. Interestingly, almost all real-world networks evolve over time by adding and removing nodes and/or edges. For instance, in social networks, people make and lose friends over time, while there are people who join the network and others who leave the network. An evolving graph is a sequence of graphs $\{\mathcal{G}_0, \mathcal{G}_1, \ldots, G_T\}$ where $\mathcal{G}_t = (A_t, X_t, E_t)$ represents the state of the evolving graph at time step $t$. It should be noted that not only nodes and edges can evolve over time, but also node and edge attributes. However, in this paper, we keep node and edge attributes fixed, and we allow only the node and edge sets of the graphs to change as a function of time. The sequence can thus be written as $\{\mathcal{G}_t = (A_t, X, E)\}_{t \in [0,T]}$. We are often interested in predicting what "comes next" in a sequence, based on data encountered in previous time steps. In our setting, this is equivalent to predicting $\mathcal{G}_t$ based on the sequence $\{\mathcal{G}_k\}_{k<t}$. In sequential modeling, we usually do not take into account the whole sequence, but only those instances within a fixed small window of size $w$ before $\mathcal{G}_t$, which we denote as $\{\mathcal{G}_{t-w}, \mathcal{G}_{t-w+1}, \ldots, \mathcal{G}_{t-1}\}$. We refer to these instances as the graph history. The problem is then to predict the topology of $\mathcal{G}_t$ given its history.

### 3.2 PROPOSED ARCHITECTURE

The proposed architecture is very similar to a typical sequence learning framework. The main difference lies in the fact that instead of vectors, in our setting, the elements of the sequence correspond to graphs. The combinatorial nature of graph-structured data increases the complexity of the problem and calls for more sophisticated architectures than the ones employed in traditional sequence learning tasks. Specifically, the proposed model consists of three components: (1) a graph neural network (GNN) which generates a vector representation for each graph instance, (2) a recurrent neural network (RNN) for sequential learning, and (3) a graph generation model for predicting the graph topology at the next time step. This framework can also be viewed as an autoencoder-decoder model. The first two components correspond to an encoder network which maps the sequence of graphs into a sequence of vectors and predicts a representation for the next in the sequence graph. The decoder network consists of the last component of the model, and transforms the above representation into a graph. Figure 1 illustrates the proposed model. In what follows, we present the different components of EvoNet in detail.

### 3.2.1 GRAPH NEURAL NETWORK

Graph Neural Networks (GNNs) have recently emerged as a dominant paradigm for performing machine learning tasks on graphs. Several GNN variants have been proposed in the past years. All these models employ some message passing procedure to update node representations. Specifically, each node updates its representation by aggregating the representations of its neighbors. After $k$

iterations of the message passing procedure, each node obtains a feature vector which captures the structural information within its $k$-hop neighborhood. Then, GNNs compute a feature vector for the entire graph using some permutation invariant readout function such as summing the representations of all the nodes of the graph. As described below, the learning process can be divided into three phases: (1) aggregation, (2) update, and (3) readout.

**Aggregation**    In this phase, the network computes a message for each node of the graph. To compute that message for a node, the network aggregates the representations of its neighbors. Formally, at time $t + 1$, a message vector $m_v^{t+1}$ is computed from the representations of the neighbors $\mathcal{N}(v)$ of $v$:

$$m_v^{t+1} = \text{AGGREGATE}^{t+1}\big(\{h_w^t \mid w \in \mathcal{N}(v)\}\big) \tag{1}$$

where AGGREGATE is a permutation invariant function. Furthermore, for the network to be end-to-end trainable, this function needs to be differentiable. In our case, AGGREGATE is a multi-layer perceptron (MLP) followed by a sum function.

**Update**    The new representation $h_v^{t+1}$ of $v$ is then computed by combining its current feature vector $h_v^t$ with the message vector $m_v^{t+1}$:

$$h_v^{t+1} = \text{UPDATE}^{t+1}\big(h_v^t, m_v^{t+1}\big) \tag{2}$$

The UPDATE function also needs to be differentiable. To combine the two feature vectors (i. e., $h_v^t$ and $m_v^{t+1}$), we have employed the Gated Recurrent Unit proposed in (Li et al., 2015):

$$h_v^{t+1} = \text{GRU}^{t+1}(h_v^t, m_v^{t+1}) \tag{3}$$

Omitting biases for readability, we have:

$$
\begin{aligned}
r_v^{t+1} &= \sigma(W_R^{t+1} m_v^{t+1} + U_R^{t+1} h_v^t) \\
z_v^{t+1} &= \sigma(W_Z^{t+1} m_v^{t+1} + U_Z^{t+1} h_v^t) \\
\tilde{h}_v^{t+1} &= \tanh(W^{t+1} m_v^{t+1} + U^{t+1}(r_v^{t+1} \odot h_v^t)) \\
h_v^{t+1} &= (1 - z_v^{t+1}) \odot h_v^t + z_v^{t+1} \odot \tilde{h}_v^{t+1}
\end{aligned}
\tag{4}
$$

where the $W$ and $U$ matrices are trainable weight matrices, $\sigma$ is the sigmoid function, and $r_v$ and $z_v$ are the parameters of the reset and update gates for a given node.

**Readout**    The *Aggregation* and *Update* steps are repeated for $T$ time steps. The emerging node representations $\{h_v^T\}_{v \in V}$ are aggregated into a single vector which corresponds to the graph representation, as follows:

$$h_G = \text{READOUT}\big(\{h_v^T \mid v \in V\}\big) \tag{5}$$

where READOUT is a differentiable and permutation invariant function. This vector captures the topology of the input graph. To generate $h_G$, we utilize *Set2Set* (Vinyals et al., 2015). Other functions such as the sum function were also considered, but were found less effective in preliminary experiments.

### 3.2.2    Recurrent Neural Networks

Given an input sequence of graphs, we use the GNN described above to generate a vector representation for each graph in the sequence. Then, to process this sequence, we use a recurrent neural network (RNN). RNNs use their internal state (i. e., memory) to preserve sequential information. These networks exhibit temporal dynamic behavior, and can find correlations between sequential events Specifically, an RNN processes the input sequence in a series of time steps (i. e., one for each element in the sequence). For a given time step $t$, the hidden state $h_t$ at that time step is updated as:

$$h_{t+1} = f(h_t, x_{t+1}) \tag{6}$$

where $f$ is a non-linear activation function. A generative RNN outputs a probability distribution over the next element of the sequence given its current state $h_t$. RNNs can be trained to predict the next element (e. g., graph) in the sequence, i. e., it can learn the conditional distribution $p(G_t|G_1, \ldots, G_{t-1})$. In our implementation, we use a Long Short-Term Memory (LSTM) network that reads sequentially the vectors $\{h_{G_i}\}_{i \in [t-w, t-1]}$ produced by the GNN, and generates a vector $h_{G_T}$ that represents the embedding of $G_T$, i. e., the graph at the next time step. The embedding incorporates topological information and will serve as input to the graph generation module. Along with the GNN component, this architecture can be seen as a form of an encoder network. This network takes as input a sequence of graphs and projects them into a low-dimensional space.

### 3.2.3 GRAPH GENERATION

To generate a graph that corresponds to the evolution of the current graph instance, we capitalize on a recently-proposed framework for learning generative models of graphs (You et al., 2018). This framework models a graph in an autoregressive manner (i. e., a sequence of additions of new nodes and edges) to capture the complex joint probability of all nodes and edges in the graph. Formally, given a node ordering $\pi$, it considers a graph $G$ as a sequence of vectors:

$$S_{\mathcal{G}}^{\pi} = (S_1^{\pi}, S_2^{\pi}, \ldots, S_{|V|}^{\pi}) \tag{7}$$

where $S_i^{\pi} = [a_{1,i}, \ldots, a_{i-1,i}] \in \{0, 1\}^{i-1}$ is the adjacency vector between node $\pi(i)$ and the nodes preceding it ($\{\pi(1), \ldots, \pi(i-1)\}$). We adapt this framework to our supervised setting.

The objective of the generative model is to maximize the likelihood of the observed graphs of the training set. Since a graph can be expressed as a sequence of adjacency vectors (given a node ordering), we can consider instead the distribution $p(\hat{S}^{\pi}; \theta)$, which can be decomposed in an autoregressive manner into the following product:

$$p(\hat{S}^{\pi}; \theta) = \prod_{i=1}^{|V|} p(\hat{S}_i^{\pi} | \hat{S}_{k:k<i}^{\pi}, \theta) = \prod_{i=1}^{|V|} \prod_{j=1}^{i-1} p(\hat{a}_{ji}^{\pi} | \hat{a}_{li:l<j}^{\pi}, \hat{S}_{k:k<i}^{\pi}, \theta) \tag{8}$$

This product can be parameterized by a neural network. Specifically, following (You et al., 2018), we use a hierarchical RNN consisting of two levels: (1) the graph-level RNN which maintains the state of the graph and generates new nodes and thus learns the distribution $p(\hat{S}_i^{\pi} | \hat{S}_{k:k<i}^{\pi})$ and (2) the edge-level RNN which generates links between each generated node and previously-generated nodes and thus learns the distribution $p(\hat{a}_{ji}^{\pi} | \hat{a}_{li:l<j}^{\pi})$. More formally, we have:

$$
\begin{aligned}
h_0 &= h_{\mathcal{G}_T} \\
h_i &= \text{RNN}_1(h_{i-1}, \hat{S}_{i-1}^{\pi}) \\
m_{0,i} &= h_i \\
m_{j,i} &= \text{RNN}_2(m_{j-1,i}, \hat{a}_{j-1,i}^{\pi}) \\
p(\hat{a}_{j,i}^{\pi} = 1) &= \sigma(m_{j,i}) \\
\hat{a}_{j,i}^{\pi} &\sim p
\end{aligned}
\tag{9}
$$

where $h_i$ is the state vector of the graph-level RNN (i. e., RNN$_1$) that encodes the current state of the graph sequence and is initialized by $h_{\mathcal{G}_T}$, the predicted embedding of the graph at the next time step $T$. The output of the graph-level RNN corresponds to the initial state of the edge-level RNN (i. e., RNN$_2$). The resulting value is then squashed by a sigmoid function to produce the probability of existence of an edge $\hat{a}_{j,i}$. In other words, the model learns the probability distribution of the existence of edges and a graph can then be sampled from this distribution, which will serve as the predicted topology for the next time step $T$.

To train the model, the cross-entropy loss between existence of each edge and its probability of existence is minimized:

$$L = \sum_{i=1}^{|V|} \sum_{j=1}^{i-1} a_{j,i}^{\pi} \big(1 - p(\hat{a}_{j,i}^{\pi} = 1)\big) + (1 - a_{j,i}^{\pi}) p(\hat{a}_{j,i}^{\pi} = 1) \tag{10}$$

**Node ordering** It should be mentioned that node ordering $\pi$ has a large impact on the efficiency of the above generative model. Note that a good ordering can help us avoid the exploration of all possible node permutations in the sample space. Different strategies such as the Breadth-First-Search ordering scheme can be employed to improve scalability (You et al., 2018). However, in our setting, the nodes are distinguishable, i. e., node $v$ of $G_i$ and node $v$ of $G_{i+1}$ correspond to the same entity. Hence, we can impose an ordering onto the nodes of the first instance of our sequence of graphs, and then utilize the same node ordering for the graphs of all subsequent time steps (we place new nodes at the end of the ordering).

## 4 EXPERIMENTS AND RESULTS

In this Section, we evaluate the performance of EvoNet on synthetic and real-world datasets for predicting the evolution of graph topology, and we compare it against several baseline methods.

### 4.1 DATASETS

We use both synthetic and real-world datasets. The synthetic datasets consist of sequences of graphs where there is a specific pattern on how each graph emerges from the previous graph instance, i. e., add/remove some graph structure at each time step. The real-world datasets correspond to single graphs whose nodes incorporate temporal information. We decompose these graphs into sequences of snapshots based on their timestamps. We fix the length of the sequences to 1000 time steps. The size of the graphs in each sequence ranges from tens of nodes to several thousand of nodes.

#### 4.1.1 SYNTHETIC DATASETS

**Path graph**    A path graph can be drawn such that all vertices and edges lie on a straight line. We denote a path graph of $n$ nodes as $P_n$. In other words, the path graph $P_n$ is a tree with two nodes of degree 1, and the other $n - 2$ nodes of degree 2. We consider two scenarios. In both cases the first graph in the sequence is $P_3$. In the first scenario, at each time step, we add one new node to the previous graph instance and we also add an edge between the new node and the last according to the previous ordering node. The second scenario follows the same pattern, however, every three steps, instead of adding a new node, we remove the first according to the previous ordering node (along with its edge).

**Cycle graph**    A cycle graph $C_n$ is a graph on $n$ nodes containing a single cycle through all the nodes. Note that if we add an edge between the first and the last node of $P_n$, we obtain $C_n$. Similar to the above case, we use $C_3$ as the first graph in the sequence, and we again consider two scenarios. In the first scenario, at each time step, we increase the size of the cycle, i. e., from $C_i$, we obtain $C_{i+1}$ by adding a new node and two edges, the first between the new node and the first according to the previous ordering node and the second between the new node and the last according to the previous ordering node. In the second scenario, every three steps, we remove the first according to the ordering node (along with its edges), and we add an edge between the second and the last according to the ordering nodes.

**Ladder graph**    The ladder graph $L_n$ is a planar graph with $2n$ vertices and $3n - 2$ edges. It is the cartesian product of two path graphs, as follows: $L_n = P_n \times P_2$. As the name indicates, the ladder graph $L_n$ can be drawn as a ladder consisting of two rails and $n$ rungs between them. We consider the following scenario: at each time step, we attach one rung ($P_2$) to the tail of the ladder (the two nodes of the rung are connected to the two last according to the ordering nodes).

For all graphs, we set the attribute of each node equal to its degree, while we set the attribute of all edges to the same value (e. g., to 1).

#### 4.1.2 REAL-WORLD DATASETS

Besides synthetic datasets, we also evaluate EvoNet on real-world datasets. These datasets contain graphs derived from the Bitcoin transaction network, a who-trust-whom network of people who trade using Bitcoin (Kumar et al., 2016; 2018). Due to the anonymity of Bitcoin users, platforms seek to maintain a record of users' reputation in Bitcoin trades to avoid fraudulent transactions. The nodes of the network represent Bitcoin users, while an edge indicates that a trade has been executed between its two endpoint users. Each edge is annotated with an integer between $-10$ and $10$, which indicates the rating of the one user given by the other user. The network data are collected separately from two platforms: Bitcoin OTC[2] and Bitcoin Alpha[3]. More details about these two datasets are given in Table 1. For all graphs, we set the attribute of each node equal to the average rating that the user has received from the rest of the community, and the attribute of each edge equal to the rating between its two endpoint users.

### 4.2 BASELINES

We compare EvoNet against several random graph models: (1) the Erdős-Rényi model (Erdős & Rényi, 1960), (2) the Stochastic Block model (Holland et al., 1983; Airoldi et al., 2008), (3) the

---

[2]https://snap.stanford.edu/data/soc-sign-bitcoin-otc.html
[3]https://snap.stanford.edu/data/soc-sign-bitcoin-alpha.html

| | $|V|$ | $|E|$ | % Pos.Edges | TIMESPAN | |
|---|---|---|---|---|---|
| | | | | BEGIN | END |
| BTC-OTC | 5, 881 | 35, 592 | 89% | 2010-11-8 | 2016-1-25 |
| BTC-ALPHA | 3, 783 | 24, 186 | 93% | 2010-11-8 | 2016-1-22 |

Table 1: Statistics of the two real-world datasets used in our experiments.

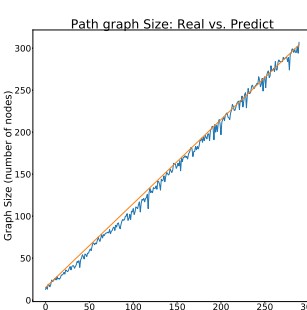 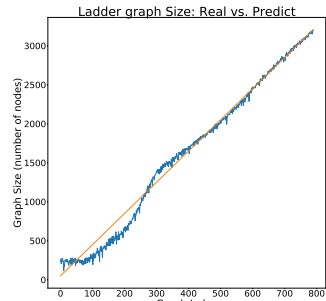 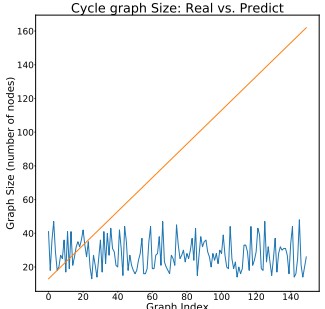

Figure 2: Results of synthetic datasets. Comparison of graph size (path, ladder and cycle graphs from left to right): predicted size (blue) VS. real size (orange).

Barabási–Albert model (Albert & Barabási, 2002), and (4) the Kronecker Graph model (Leskovec et al., 2010). These are the traditional methods to study the topology evolution of temporal graphs, by proposing a driven mechanism behind the evolution.

### 4.3 EVALUATION METRIC AND EXPERIMENTAL SETUP

In general, it is very challenging to measure the performance of a graph generative model since it requires comparing two graphs to each other, a long-standing problem in mathematics and computer science (Conte et al., 2004). We propose to use graph kernels to compare graphs to each other, and thus to evaluate the quality of the generated graphs. Graph kernels have emerged as one of the most effective tools for graph comparison in recent years (Nikolentzos et al., 2019). A graph kernel is a symmetric positive semidefinite function which takes two graphs as input, and measures their similarity. In our experiments, we employ the Weisfeiler-Lehman subtree kernel which counts label-based subtree-patterns (Shervashidze et al., 2011). Note that we also normalize the kernel values, and thus the emerging values lie between 0 and 1.

As previously mentioned, each dataset corresponds to a sequence of graphs where each sequence represents the evolution of the topology of a single graph in 1000 time steps. We use the first 80% of these graph instances for training and the rest of them serve as our test set. The window size $w$ is set equal to 10, which means that we feed 10 consecutive graph instances to the model and predict the topology of the instance that directly follows the last of these 10 input instances. Each graph of the test set along with its corresponding predicted graph is then passed on to the Weisfeiler-Lehman subtree kernel which measures their similarity and thus the performance of the model.

The hyperparameters of EvoNet are chosen based on its performance on a validation set. The parameters of the random graph models are set under the principle that the generated graphs need to share similar properties with the ground-truth graphs. For instance, in the case of the Erdős-Rényi model, the probability of adding an edge between two nodes is set to some value such that the density of the generated graph is identical to that of the ground-truth graph. However, since the model should not have access to such information (e. g., density of the ground-truth graph), we use an MLP to predict this property based on past data (i. e., the number of nodes and edges of the previous graph instances). This is in par with how the proposed model computes the size of the graphs to be generated (i. e., using also an MLP).

### 4.4 RESULTS

**Synthetic datasets** Figure 2 illustrates the experimental results on the synthetic datasets. Since the graph structures contained in the synthetic datasets are fairly simple, it is easy for the model to

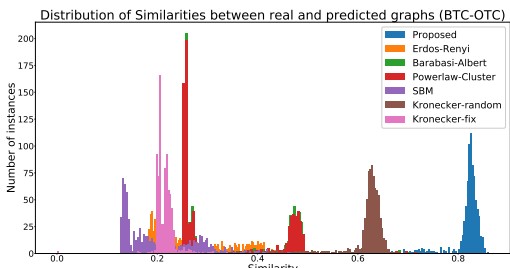 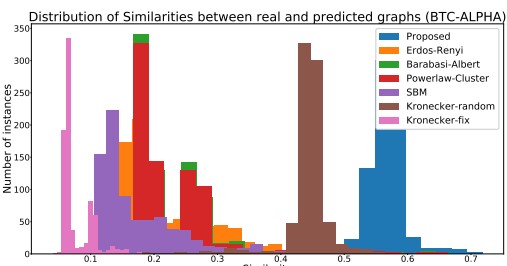

Figure 3: Similarity histograms on BTC-OTC (left) and BTC-Alpha (right) datasets. Blue one is the result of EvoNet, which is compared against 6 random graph models.

| STAT.
MODEL | BTC-OTC | | BTC-ALPHA | |
|---|---|---|---|---|
| | MEAN | 90%ILE | MEAN | 90%ILE |
| ER | 0.28 | 0.40 | 0.22 | 0.32 |
| SBM | 0.21 | 0.30 | 0.18 | 0.27 |
| BA | 0.35 | 0.48 | 0.23 | 0.28 |
| POWER | 0.35 | 0.48 | 0.23 | 0.28 |
| KRON-RAND | 0.62 | 0.64 | 0.44 | 0.47 |
| KRON-FIX | 0.21 | 0.23 | 0.08 | 0.11 |
| EVONET | **0.82** | **0.84** | **0.55** | **0.59** |

Table 2: Statistics on the similarity distribution of different models: ER stands for Erdős–Rényi model. SBM stands for Stochastic Block Model. BA is Barabási–Albert Model. POWER is another model, similar to the Barabási–Albert, that grows graphs with powerlaw degree distribution. Kron-Rand represents the Kronecker Graph Model with learnable parameter while Kron-Fix represents the Kronecker Graph Model with fixed parameters which depend on the initial graph.

generate graphs very similar to the ground-truth graphs (normalized kernel values $> 0.9$). Hence, instead of reporting the kernel values, we compare the size of the predicted graphs against that of the ground-truth graphs. The figures visualize the increase of graph size on real sequence (orange) and predicted sequence (blue). For path graphs, in spite of small variance, we have an accurate prediction on the graph size. For ladder graph, we observe a mismatch at the beginning of the sequence for small size graphs but then a coincidence of the two lines on large size graphs. This mismatch on small graphs may be due to a more complex structure in ladder graphs such as cycles, as supported by the results of cycle graph on the right figure, where we completely mispredict the size of cycle graphs. In fact, we fail to reconstruct the cycle structure in the prediction, with all the predicted graphs being path graphs. This failure could be related to the limitations of GNN mentioned in Xu et al. (2018)

**Real-World datasets** Finally, we analyze the performance of our model on real datasets: the Bitcoin-OTC and Bitcoin-Alpha. We obtain the similarities between each pair of real and predicted graphs in the sequence and draw a histogram to illustrate the distribution of similarities. The results are shown in Figure 3 respectively for the two datasets. Among all the traditional random graph models, Kronecker graph model (with learnable parameter) performs the best, however on both datasets, our proposed method EvoNet (in blue) outperforms tremendously all other methods, with an average similarity of $0.82$ on BTC-OTC dataset and $0.55$ on BTC-Alpha dataset. Detailed statistics can be found in Table 2.

Overall, our experiments demonstrate the advantage of EvoNet over the traditional random graph models on predicting the evolution of dynamic graphs.

## 5 CONCLUSION

In this paper, we proposed EvoNet, a model that predicts the evolution of dynamic graphs, following an encoder-decoder framework. We also proposed an evaluation methodology for this task which capitalizes on the well-established family of graph kernels. Experiments show that the proposed model outperforms traditional random graph methods on both synthetic and real-world datasets.

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

## A    EXTRA EXPERIMENT RESULTS WITH SYNTHETIC DATASETS

### A.1    GRAPH SIZE COMPARISON

See Figure 4.

### A.2    HISTOGRAM OF SIMILARITIES

See Figure 5.

### A.3    SOME EXAMPLES OF PREDICTED GRAPHS

See Figure 6, 7, 8, 9, 10, 11, respectively for Path graphs, Ladder graphs with small size, Ladder graphs with large size, Cycle graphs, Path graphs with removal, Cycle graphs with adding extra structures.

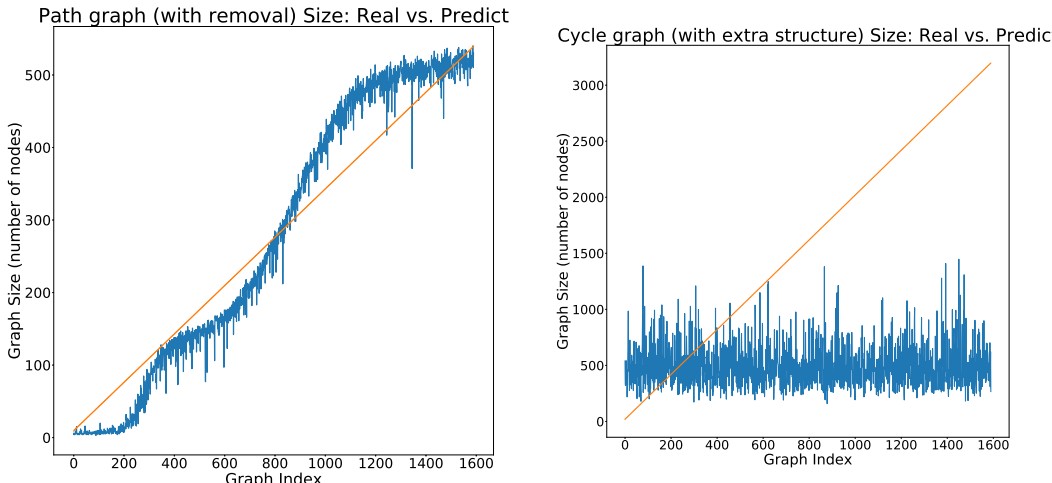

Figure 4: Comparison of graph size: predicted size (blue) VS. real size (orange). Left: Path graphs with removal; Right: Cycle graphs with adding extra structures

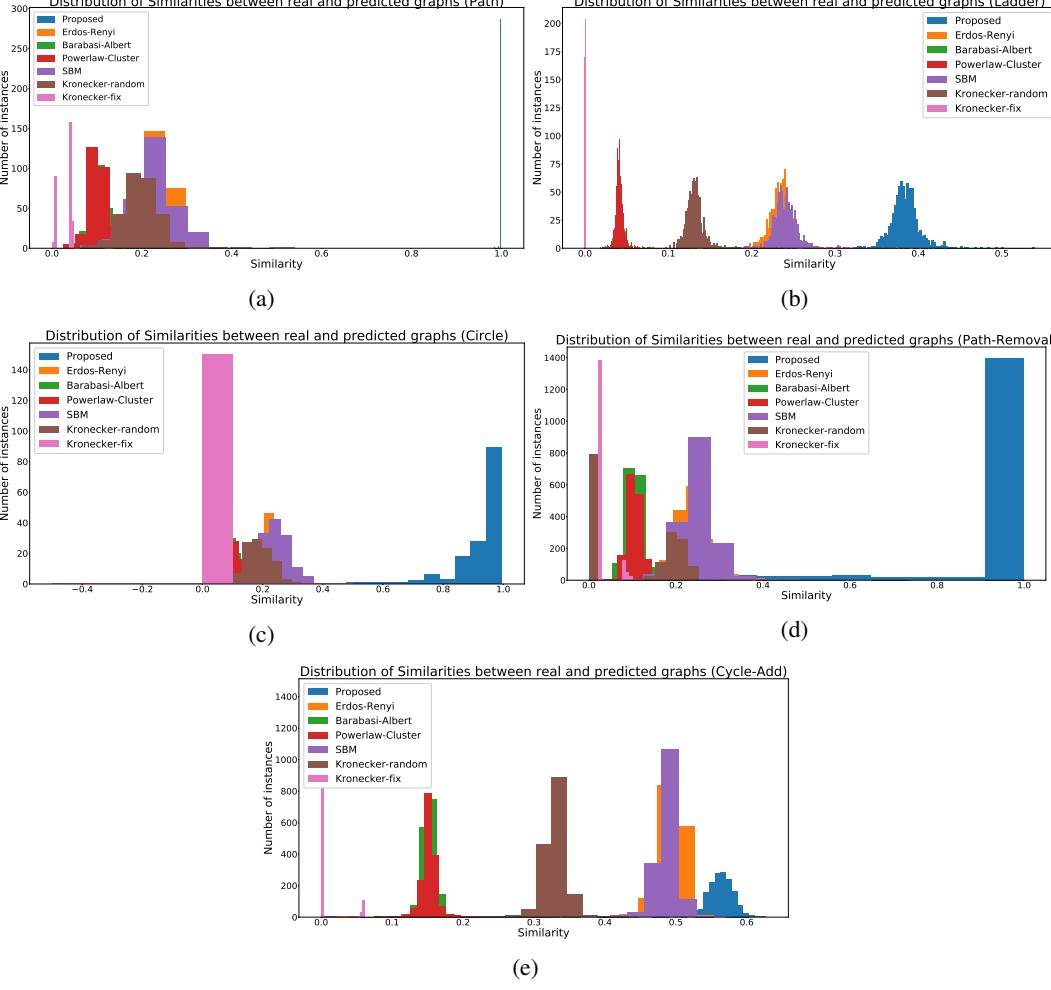

Figure 5: Similarity histograms on synthetic datasets. Blue one is the result of EvoNet, which is compared against 6 random graph models. 5a: Path graphs; 5b: Ladder graphs; 5c: Cycle graphs; 5d: Path graphs with removal; 5e: Cycle graphs with adding extra structures.

Real graphs          Predicted graphs

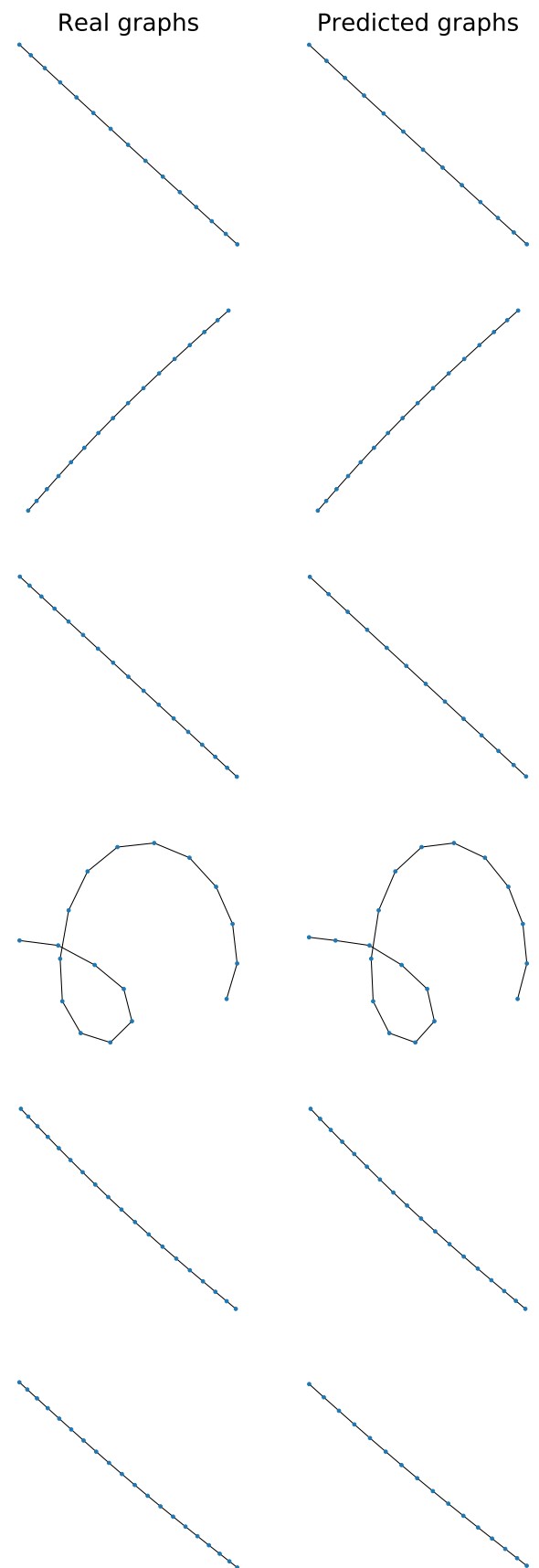

Figure 6: Some examples of predictions on Path datasets: the left column is the real graphs and the right column is the predicted ones.

Real graphs        Predicted graphs

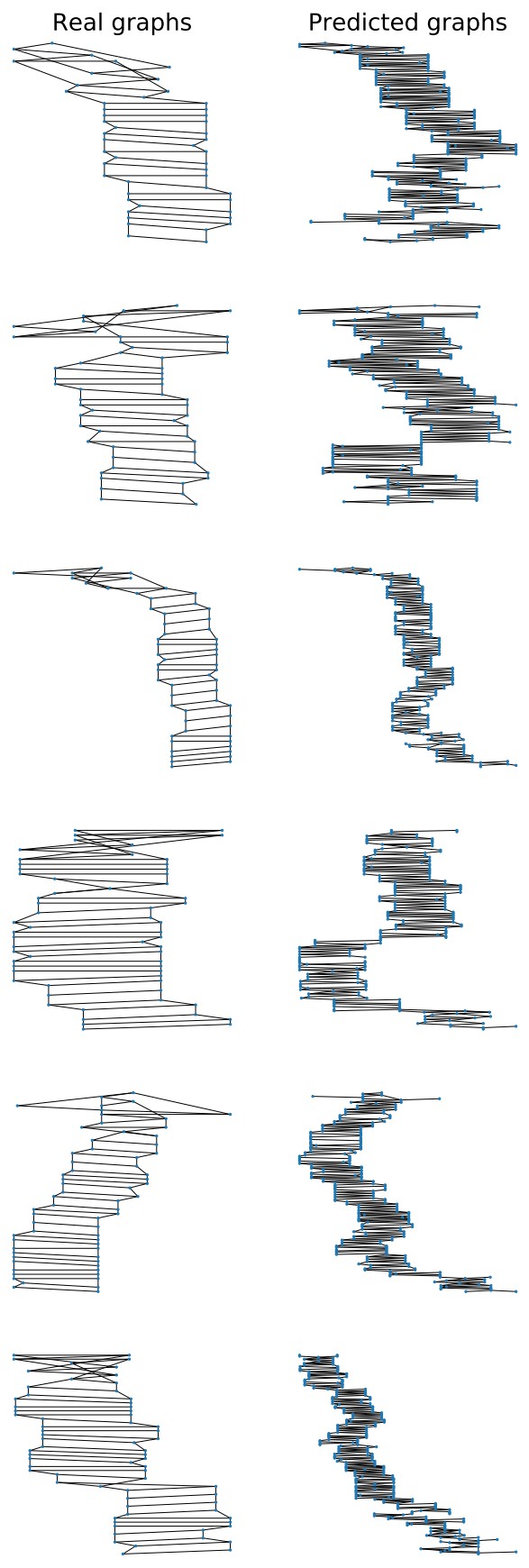

Figure 7: Some examples of predictions on Ladder datasets (with small size graphs): the left column is the real graphs and the right column is the predicted ones.

Real graphs       Predicted graphs

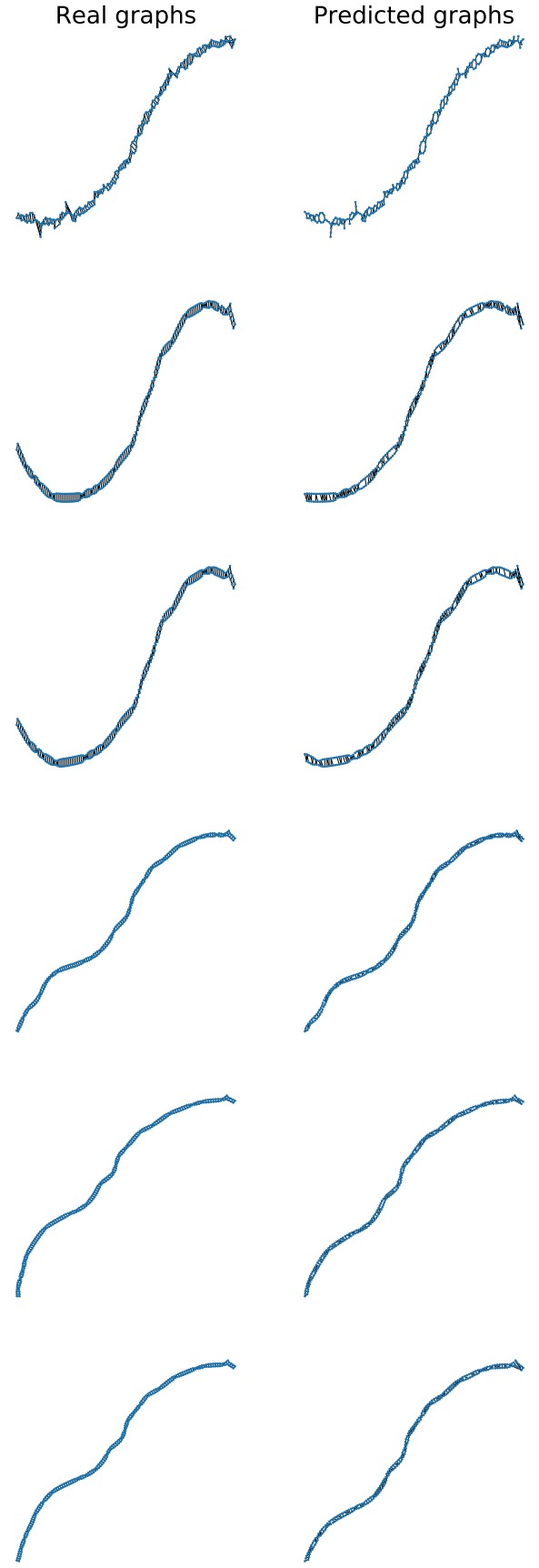

Figure 8: Some examples of predictions on Ladder datasets (with large size graphs): the left column is the real graphs and the right column is the predicted ones.

Real graphs          Predicted graphs

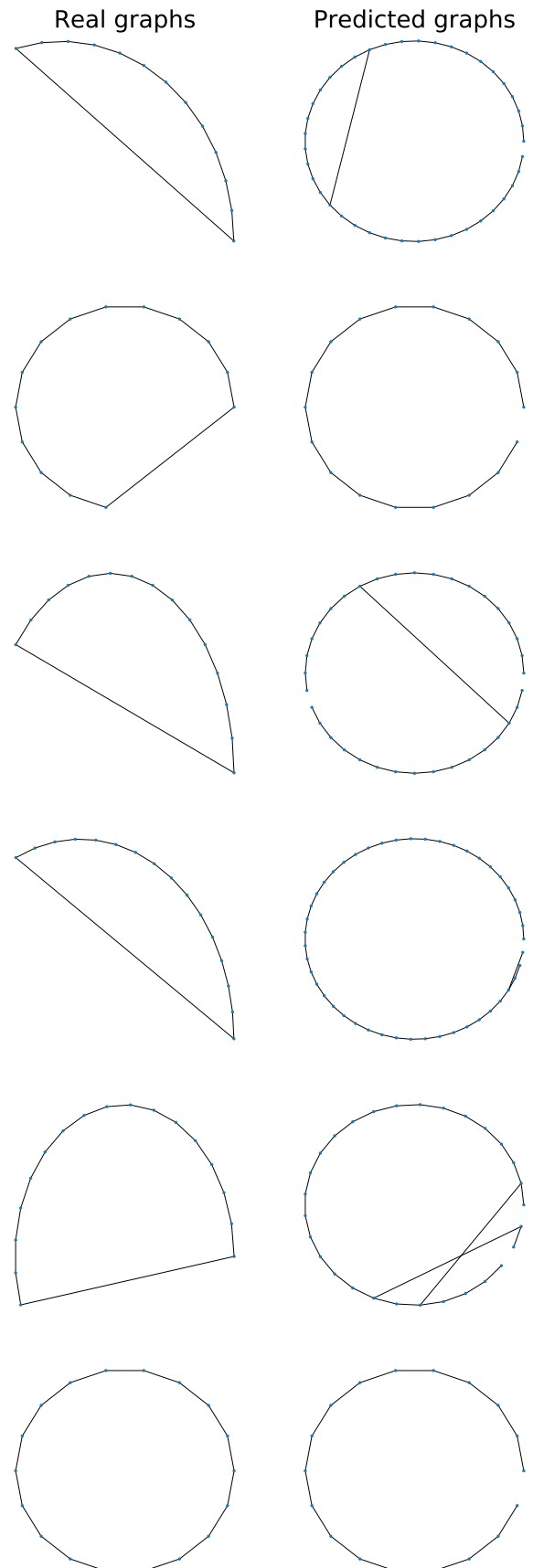

Figure 9: Some examples of predictions on Cycle datasets: the left column is the real graphs and the right column is the predicted ones.

Real graphs          Predicted graphs

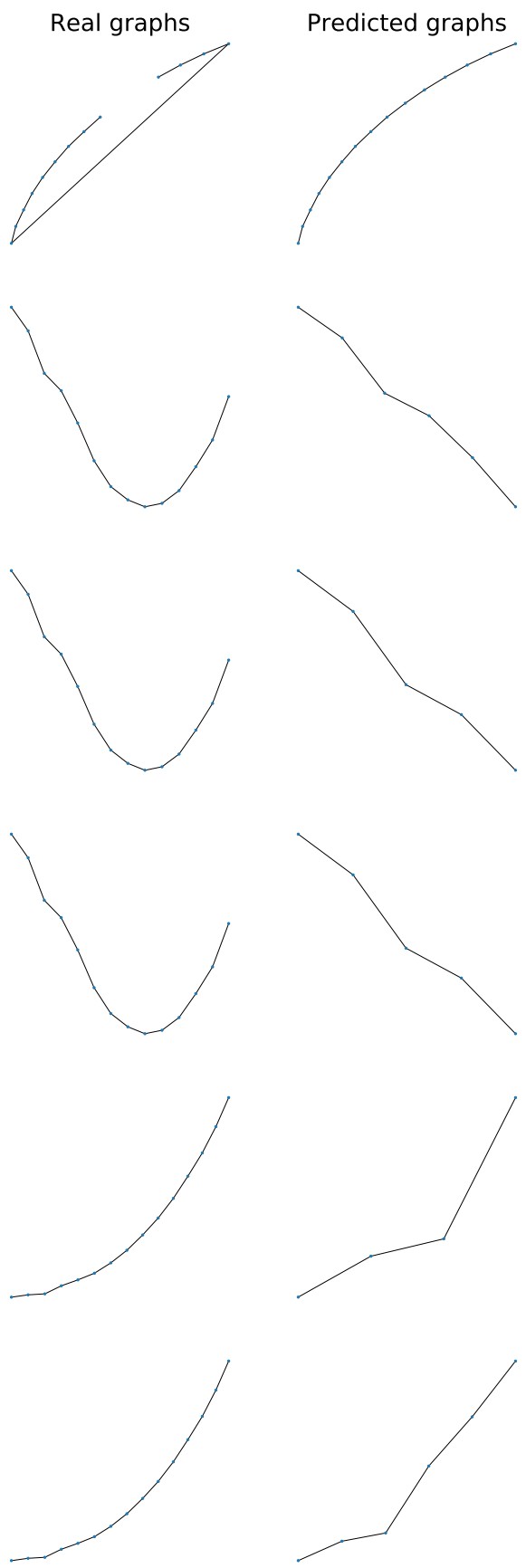

Figure 10: Some examples of predictions on Path datasets with removal: the left column is the real graphs and the right column is the predicted ones.

Real graphs          Predicted graphs

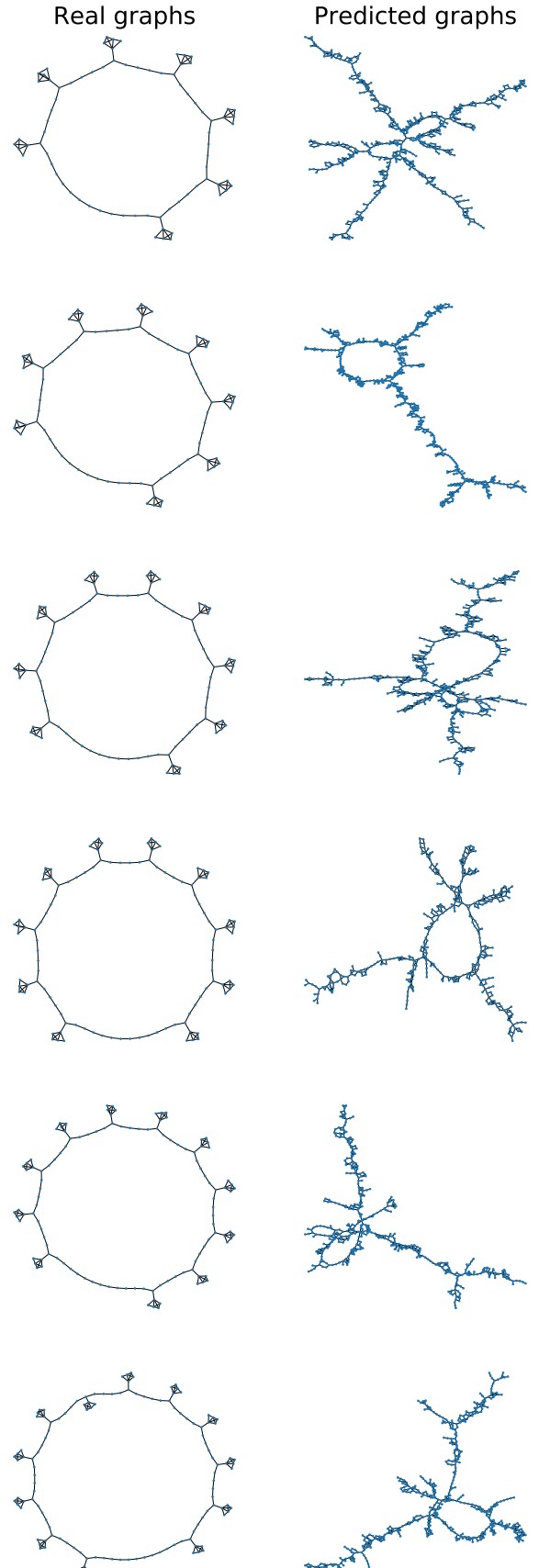

Figure 11: Some examples of predictions on Cycle datasets with adding extra structures: the left column is the real graphs and the right column is the predicted ones.

