# OpenReview forum: "EvoNet: A Neural Network for Predicting the Evolution of Dynamic Graphs"
_ICLR.cc/2020/Conference — Reject_

### Official Review · AnonReviewer1 · 2019-10-23
**Official Blind Review #1**

**Rating:** 3

**Review:**

This paper presents a system for predicting evolution of graphs. It makes use of three different known components - (a) Graph Neural Networks (GNN); (b) Recurrent Neural Networks (RNN); (c) Graph Generator. A significant portion of the paper is spent in explaining these known concepts. The contribution of the paper seems to be a system of combining these to achieve graph evolution prediction. As stated, this system is effectively a recurrent auto-encoder of sorts.

The main objection I have in this paper is that they have only used two real datasets (both of which are from the same domain). There are several only available datasets that have temporally annotated graph evolution. It is not possible to conclude the empirical superiority of a system based on such little evidence.

**Experience Assessment:**

I have read many papers in this area.

**Review Assessment: Checking Correctness Of Derivations And Theory:**

I assessed the sensibility of the derivations and theory.

**Review Assessment: Checking Correctness Of Experiments:**

I carefully checked the experiments.

**Review Assessment: Thoroughness In Paper Reading:**

I read the paper at least twice and used my best judgement in assessing the paper.

---

### Official Review · AnonReviewer3 · 2019-10-23
**Official Blind Review #3**

**Rating:** 3

**Review:**

In this paper, the authors propose a new neural network architecture for predicting the next graph conditioned on a past graph sequence. It seems that the proposed model is the first deep learning model for graph sequence prediction. The model consists of three major components: a graph encoder that maps a graph to an encoding represented as a vector, an LSTM for graph sequence embedding, and a graph decoder for generating an affinity matrix.

There are two main concerns I have with this paper:
- The model has some inherent limitations in the graph embedding step. First, the graph encoder embeds a graph into a feature vector that represents the topology of the graph. I assume that the feature vector has a small size, and it is hard to encode a large graph (i.e. 1000x1000). This representation is quite sub-optimal to me. The model will not be able to utilize the complete information of a large dense graph.  Second, the model only takes 10 graphs as input and ignores other graphs in the input graph sequences. This sounds suboptimal to me.

- The performance of the proposed model is not satisfactory. The model does not output a graph with the right size for very simple synthetic graphs. The model completely fails for generating circles. A better model should be proposed to address this challenge. Evaluation is not convincing enough. Simply comparing the graph size between the output and ground truth is not sufficient. We can further predict where graph structure matches the ground truth exactly. I believe this can be done for simple graphs like circles, paths, and ladders.

Other comments:
- The authors claim that all the sequences in the datasets are fixed to 1000. However, in Figure 4, the graph index goes up to 1600.  Why?


**Experience Assessment:**

I have published one or two papers in this area.

**Review Assessment: Checking Correctness Of Derivations And Theory:**

I assessed the sensibility of the derivations and theory.

**Review Assessment: Checking Correctness Of Experiments:**

I assessed the sensibility of the experiments.

**Review Assessment: Thoroughness In Paper Reading:**

I read the paper at least twice and used my best judgement in assessing the paper.

---

### Official Review · AnonReviewer2 · 2019-10-23
**Official Blind Review #2**

**Rating:** 1

**Review:**

This paper proposes a framework to model the evolution of dynamic graphs for the task of predicting the topology of next graph given a sequence of graphs. Specifically, the paper uses a combination of recently proposed techniques in graph representation learning (Graph Neural Network) and Graph Generation (GraphRNN [You et. al. 2018]). Given a sequence of graphs as input, a GNN (to obtain low-dimensional representations of the graphs in this sequence) and LSTM (to model the sequence of these representations)  based encoder is used to compute a vector representation of the topology of next graph in the sequence. The learned vector is then used as input to a GraphRNN decoder to generate a graph that would serve as a predicted next graph in the sequence. The proposed approach is validated with experiments on three synthetic datasets and one real-world dataset (Bitcoin is same dataset from two different resources with little difference in characteristics) and compared against random graph models.

This paper should be rejected due to following reasons:
(1) The authors do not justify/discuss the motivation and importance of the task and corresponding applications that would require to predict topology of complete graph in the next step.
(2) The proposed techniques are an adhoc combination of existing techniques with major concerns (details below) but also with little novelty (if any) for achieving this combination.
(3) The empirical efforts are very limited and does not provide enough evidence about the efficacy of the method, miss several details and does not serve as motivation for designing such a method in first place. Please note that negative results on cycle graphs has no role to play in this assessment. In fact,
I appreciate the authors for reporting negative results as it provides a transparent insights into the effectiveness of model in different settings.
Overall, the paper needs lot of work on all aspects - motivation, technique and experiments to make it fit for a conference publication.

Major Concerns:

(a) Motivation: The authors do not discuss or motivate the problem and why it is important to the community. The authors mention that many existing work on dynamic graphs focus on learning representations. This is the case because learned representations can then be used for various downstream applications and even future event predictions. When such methods can be used to do future predictions required for most applications, why does one need to predict the topology of complete next graph? The authors need to provide concrete justification for the problem they address, instances where such a task would be useful and discussion on other techniques that can do similar tasks but lack in aspects that such a method can capture. For instance, as a preliminary step, can the authors explain how solving this problem would be helpful to bitcoin?

(b) Technical: The technical contributions of this paper lack novelty and has several flows:

- Figure 1 seems to show that graph only grows in size. While the authors do provide an experiment with removal process, that experiment does not seem to perform well. So, does the method is only good to support growing graphs?
- Authors mention that the edge and node attributes are considered to be fixed. However, if the number of nodes and edges change, X and L should also change in terms of dimensions and adding values for new nodes/edges. so why should it not be considered time-varying?
- What was the motivation for using GRU for update function in Eq 3? Was simple MLP tried and not useful? Was GRU used to capture some long term dependencies in structure? If so, the authors must explain how it is useful for this task.
- Why Set2Set was used for ReadOut function? This seems to be a particularly adhoc and odd choice. when sum did not work well, jump to Set2Set is not justified. Can the authors provide an explanation for the same?
- The authors claim that the embedding h_G_T incorporates topological information -- I find this claim highly unsubstantiated and needs justification. For instance, can you provide some rigorous analysis to demonstrate that this is the case? At the least, can the authors use this vector and pass it through a graph decoder to recover the original graph?
- What is novel in 3.2.3 as compared to You et. al.? Infact, it is hard to see any novelty in the entire combination. Was it challenging to achieve this combination? If so, what was the challenging part? It is not clear what the authors contributed to address such a challenge. Was the training challenging? If so, please explain. If not, why is this a novel approach?

(c) Empirical: The empirical efforts are inadequate and raises more questions than answers.

- Synthetic datasets are simple and more datasets should be used e.g. You et. al. 2018 to validate the performance. Only one real-world dataset from two different sources is used. It is hard to understand author's motivation in doing so. Why not use various graph datasets available in papers that learn representations (e.g. cited by authors themselves) What is special about bitcoin dataset that makes it suitable for this task?

- Node/edge attributes are chosen in adhoc manner and it is unclear what role they perform. Do they help with prediction? If not, would it be useful to first show experiments without them? Or does this method absolutely need attributes? It is not clear why it is useful to set all attributes for edge as 1.

- How was window size of 10 chosen? Why is the same window size good for all graphs? What impact does window size has on performance?

- What is the motivation for using Graph kernel for similarity? The authors borrow the decoder from You et. al. 2018 which also provides a principled method to compare graphs using MMD based on statistics. Why not employ the same?

- GraphRNN (You et. al.) and other generative models can learn over multiple graphs? Did the authors try to feed the sequence of graphs to such models and then try to generate a new graph to see if they can produce similar results? It is true that those generative models do not specifically model temporal sequence, but such an experiment would help to distinguish the efficacy of the proposed method.

-The technique of using MLP for generating predictions using random graph models seem to be highly unfair for the baselines. Can you elaborate more as it is difficult to understand why one should handicap those models by using learned information instead of data information?

- A rigorous discussion on insights explaining the results is required. The authors show high performance on Bitcoin dataset. However,  it is not clear what part is contributing to the performance. Similarly, authors should dig deep into the failure cases and provide justification of why such a method would fail in particular cases and propose alternatives.

- Why was Graph size used as a statistic to report? Two graphs of same size can be entirely different and I do not see any merit in using such a metric. Again, something like MMD based metric may be useful.


Improvements that would make future revision strong but has not impacted current assessment:

Overall, the presentation of the paper is very unpolished. The authors are missing many important details as described above while spending a lot of time in describing (repeating) known techniques verbatim as original works. This can be removed and condensed into very short preliminary section.

- Notations: The authors must use clear notations. For instance, on Page 2, L is used to  describe edge attributes but then it is replaced by E in Page 3. Also, both X and L are shown to have dimension d. Are edge and node attributes of same dimension? w is used for window-size of sequence used as input and also as neighbor node. When modeling evolution of graphs where a sequence is available over time points 0...T, it is not useful to use T to also represent time step of GNN propagation. Infact, authors should avoid using time steps to signify GNN iterations.

- Empirical details: The details provided for datasets and experimental setup is inadequate. Why are the two Bitcoin datasets different from each other? What does Pos. Edges in Table 1 mean? What does  Mean and 90th percentile in Table 2 signify? Authors only talk about train-test split but then mention
validation set for hyper-param tuning. How was this validation set obtained? Also, what hyper-params were tuned and what was sensitivity of those hyper-params? Authors use GNN and multiple RNN's, what was the model capacity used and how it impacted the performance? Figure 5 (c) what is a circle graph?

**Experience Assessment:**

I have published in this field for several years.

**Review Assessment: Checking Correctness Of Derivations And Theory:**

N/A

**Review Assessment: Checking Correctness Of Experiments:**

I carefully checked the experiments.

**Review Assessment: Thoroughness In Paper Reading:**

I read the paper thoroughly.

---

### Decision · Program_Chairs · 2019-12-19

**Decision:**

Reject

**Comment:**

The paper proposes a combination graph neural networks and graph generation model (GraphRNN) to model the evolution of dynamic graphs for predicting the topology of next graph given a sequence of graphs.

The problem to be addressed seems interesting, but lacks strong motivation. Therefore it would be better if some important applications can be specified.

The proposed approach lacks novelty. It would be better to point out why the specific combination of two existing models is the most appropriate approach to address the task.

The experiments are not fully convincing. Bigger and comprehensive datasets (with the right motivating applications) should be used to test the effectiveness of the proposed model.

In short, the current version failed to raise excitement from readers due to the reasons above. A major revision addressing these issues could lead to a strong publication in the future.